# Molecular evolution and the decline of purifying selection with age

Changde Cheng[1] & Mark Kirkpatrick [2✉]

Life history theory predicts that the intensity of selection declines with age, and this trend should impact how genes expressed at different ages evolve. Here we find consistent relationships between a gene's age of expression and patterns of molecular evolution in two mammals (the human *Homo sapiens* and the mouse *Mus musculus*) and two insects (the malaria mosquito *Anopheles gambiae* and the fruit fly *Drosophila melanogaster*). When expressed later in life, genes fix nonsynonymous mutations more frequently, are more polymorphic for nonsynonymous mutations, and have shorter evolutionary lifespans, relative to those expressed early. The latter pattern is explained by a simple evolutionary model. Further, early-expressed genes tend to be enriched in similar gene ontology terms across species, while late-expressed genes show no such consistency. In humans, late-expressed genes are more likely to be linked to cancer and to segregate for dominant disease-causing mutations. Last, the effective strength of selection ($N_e\ s$) decreases and the fraction of beneficial mutations increases with a gene's age of expression. These results are consistent with the diminishing efficacy of purifying selection with age, as proposed by Medawar's classic hypothesis for the evolution of senescence, and provide links between life history theory and molecular evolution.

[1] Department of Computational Biology, St. Jude Children's Research Hospital, Memphis, TN, USA. [2] Department of Integrative Biology, University of Texas, Austin, TX, USA. ✉email: kirkp@mail.utexas.edu

In stable populations, the force of selection declines with age as the number of individuals in a cohort that are subject to selection is thinned by mortality. Consequently, one expects that genes most highly expressed late in life will tend to experience weaker purifying selection than those expressed early in life. This effect should cause late expressed genes to show the increased accumulation of deleterious mutations. This fundamental principle was first articulated by Medawar[1,2] (later made rigorous by Hamilton[3] and Charlesworth[4]) and is one of the leading hypotheses for the evolution of senescence[4].

Several recent studies have found support for this idea in patterns of molecular evolution. Rodriguez et al.[5,6] showed that genes in humans associated with late-onset diseases have higher minor allele frequencies and larger effect sizes than those associated with early-onset diseases. Jia et al.[7] reported that late-expressed genes in humans have higher rates of nonsynonymous substitution ($d_N/d_S$) and are younger. Turan et al.[8] also showed in four additional mammal species that late-expressed genes have higher values of $d_N/d_S$. As most nonsynonymous mutations are deleterious while most synonymous mutations evolve neutrally[9], these results are consistent with Medawar's argument.

Here, we expand the phylogenetic scope and the number of molecular signatures that are correlated with the ages at which a gene is expressed. Data from two mammals (the human *Homo sapiens* and the mouse *Mus musculus*) and two insects (the fruit fly *Drosophila melanogaster* and the malaria mosquito *Anopheles gambiae*) show several repeatable patterns. First, late expressed genes ("late genes") have higher rates of nonsynonymous substitution ($d_N/d_S$) than early expressed genes ("early genes"). Second, late genes are more polymorphic than early genes for nonsynonymous mutations. Third, late genes are on average younger than early genes. These first three results are mirrored in comparisons of pairs of paralogs that are expressed at different ages. Fourth, early genes are consistently enriched in certain gene ontology categories, while late genes are rarely enriched in any category. In humans, late genes are enriched for loci linked to cancer and loci that segregate for dominant disease-causing mutations. Last, late genes tend to experience weaker effective strengths of selection ($N_e \, s$). With the support of two simple models, we interpret these patterns as resulting from a decline in the force of purifying selection with age.

## Results

**Age of expression vs. polymorphism, substitution rate, and gene age.** We quantified how a gene's expression varies across the lifespan using the regression of expression on age, or REA. A positive value of REA means the gene is expressed most strongly late in life, while a negative value means expression is strongest early. Expression was quantified using the log of read counts, therefore differences are relative. We asked how three characteristics of genes depend on REA: $d_N/d_S$, which is the ratio of the rates of nonsynonymous and synonymous substitutions; $p_N/p_S$, which is the ratio of the fraction of nonsynonymous sites that are polymorphic to the fraction of synonymous sites that are polymorphic, and the age of a gene. We regressed each of these statistics onto REA, and as covariates we controlled for the mean levels of expression and the breadth of tissues in which genes are expressed (see "Methods" section).

We expect that the declining force of purifying selection with age will cause $d_N/d_S$ to be positively correlated with REA. That pattern is seen in all four species (Fig. 1 and Supplementary Fig. 1). (Values of the Spearman's rank correlation for these relations and those that follow are given in Supplementary Table 1.) This trend could also result from other causes, for

example if positive selection is much more common in late than early genes, but there does not seem to be a compelling biological reason for that to be so.

Another pattern is evident in the data: the relative rates that $d_N/d_S$ changes with REA are correlated with the average synonymous diversity, $\bar{\pi}_S$. To quantify this relation, we calculated $\Delta R_{NS}$, defined as the proportional difference in the $d_N/d_S$ ratio between early genes and late genes (see "Methods" section). The results are shown in Table 1 and Supplementary Table 1. The relative changes in $d_N/d_S$ with REA are much greater in the two insects than in the two mammals, and the insects have higher average synonymous diversity. A simple hypothesis to explain this relation is that vertebrates have smaller effective population sizes than insects, which is reflected by lower values of $\bar{\pi}_S$. This causes purifying selection to be less efficient overall, which in turn diminishes the effect that age has on the strength of purifying selection. While suggestive, these data do not allow a strong test of that hypothesis because the number of data points (four species) is too small for statistical significance and the species are not phylogenetically independent.

We expect nonsynonymous mutations to persist at higher frequencies at loci that are mainly expressed late in life. Figure 2 and Supplementary Fig. 2 show that $p_N/p_S$ does indeed increase significantly with REA in all four species. Further, the effects of REA are large: the average value of $p_N/p_S$ increases between 12% (in humans) to over 125% (in the mosquito) for late genes compared to early genes. For all four species, the relation between $p_N/p_S$ and REA remains significant using only data from autosomes.

Next, we considered how the evolutionary lifespan of a gene varies with the ages at which it is most strongly expressed. Figure 3 and Supplementary Fig. 3 show that gene age declines with REA, which implies that late genes have shorter evolutionary lives than early genes. A hypothesis to explain this pattern is that genes are more likely to be lost when they become sufficiently degraded by the fixation of deleterious mutations, and this happens more frequently to late genes because they experience weaker purifying selection. The "Methods" section presents a highly simplified model that corroborates this intuition. The model assumes that genes fix a sequence of beneficial and deleterious mutations, which causes their fitness to evolve as a random walk. A gene is lost (deleted or pseudogenized) if fixation of deleterious mutations causes its fitness to decline past a certain threshold. Weaker purifying selection causes deleterious mutations to become fixed more frequently. Under those conditions, late genes (which experience weaker purifying selection) will be lost more quickly, so that class of genes will tend to be younger than early genes.

Three alternative hypotheses might explain the relation between when a gene is expressed and its age. First, genes expressed early in development often contribute to basic features of the body plan that are conserved over long spans of evolutionary time, so these genes might experience positive selection less often. This explanation seems unlikely, however, as our data come from later stages of the life cycle (e.g., adults in the cases of the two insects).

Second, because late genes are evolving faster (Fig. 1), they may more often diverge enough from homologs in other species that they are identified as new genes, for example because they experience relaxed purifying selection. To address this hypothesis, we considered pairs of paralogs that are expressed at different ages. For each pair, we regressed $d_N/d_S$, $p_N/p_S$, and gene age onto REA, with mean expression level and tissue specificity as covariates. In all four species, $d_N/d_S$ is significantly greater and gene age is significantly younger, in genes that are expressed late

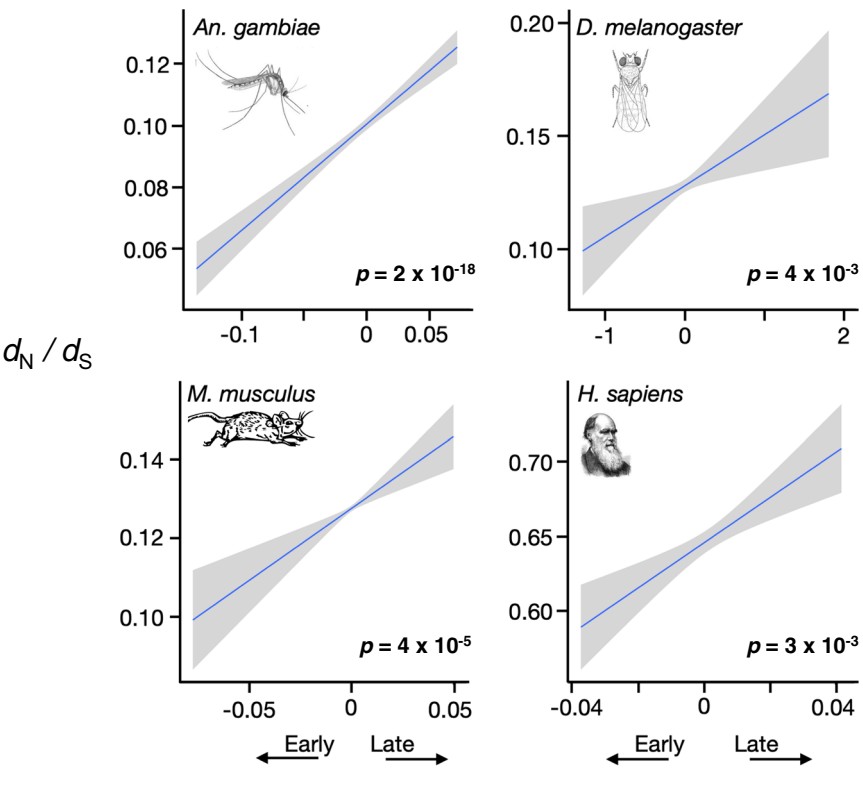

**Fig. 1 Late genes fix nonsynonymous mutations at higher rates.** Genes expressed later in life fix nonsynonymous mutations at a significantly higher relative rates than those expressed early, as measured by the $d_N/d_S$ ratio, in mosquitoes, flies, mice, and humans. Significance was determined by Spearman rank correlations, and p-values in bold are significant at $p < 0.05$ (two-sided tests, not corrected for multiple tests). The lines are least squares regressions, and the gray regions show the approximate 95% confidence intervals for the regressions. See also Supplementary Fig. 1 and Supplementary Table 1.

**Table 1 The proportional change in the relative fixation rate of nonsynonymous mutations in early vs. late genes, $\Delta R_{NS}$, is positively correlated with average diversity at synonymous sites, $\bar{\pi}_S$, across the four species.**

| Species | $\bar{\pi}_S$ | $\Delta R_{NS}$ |
|---|---|---|
| Mosquito | 0.0072 | 0.58 |
| Fruit fly | 0.0035 | 0.09 |
| Mouse | 0.0016 | 0.03 |
| Human | 0.0008 | 0.02 |

compared to their paralogs that are expressed early (Table 2). The trend is in the same direction for $p_N/p_S$, but is significant in only two of the four species. These results are inconsistent with the alternative hypothesis, i.e., that late genes diverge more often to the point that they are no longer identified as homologous across species, but are consistent with decreasing strength of purifying selection with age.

A third alternative hypothesis is that genes expressed at different ages tend to fall into different functional classes, and these classes experience different intensities of purifying selection (for some unknown reason). To test this hypothesis, we asked if genes expressed early or late differ with respect to their gene ontology (GO) categories. In the four species, between 11 and 1692 GO categories are significantly enriched among early genes (Supplementary Table 2). Further, some categories are shared between species. Among the ten most

significant terms for each species, ribosome biogenesis appears in three species, and four other terms (helicase activity, nucleolus, ribosome, and translation) are shared between two species. The results for late genes are very different. Only four GO terms in the mosquito are significantly enriched, and no terms are enriched in the other three species (Supplementary Table 2). This result is consistent with the third alternative hypothesis and also with the diminishing intensity of purifying selection (see the "Discussion" section).

**Age of expression and association with cancer**. In humans, we also find that late genes are much more likely than early genes to be driver genes for adult cancers (Fig. 4 and Supplementary Fig. 4). The probability of being a driver is roughly 150% greater for the latest-expressed genes than the earliest-expressed genes. This is consistent with the fixation of deleterious mutations leading to the misregulation of late genes. Likewise, we found that late genes are significantly enriched for loci that segregate for dominant disease-causing mutations, relative to early genes ($p < 0.02$, Wilcoxon rank-sum test).

**Age of expression and fitness effects of new mutations**. Next we asked if the effective strength of selection, measured as $N_e s$, and the fraction of new nonsynonymous mutations that are beneficial, denoted as $\alpha_m$, varies with the ages at which genes are expressed. To do so, we developed a very simple model (see "Methods" section). It assumes that in an equilibrium population with effective size $N_e$, mutations occur at a total rate of $\mu_N$ across all the nonsynonymous sites in a gene, and at a rate $\mu_S$ across the

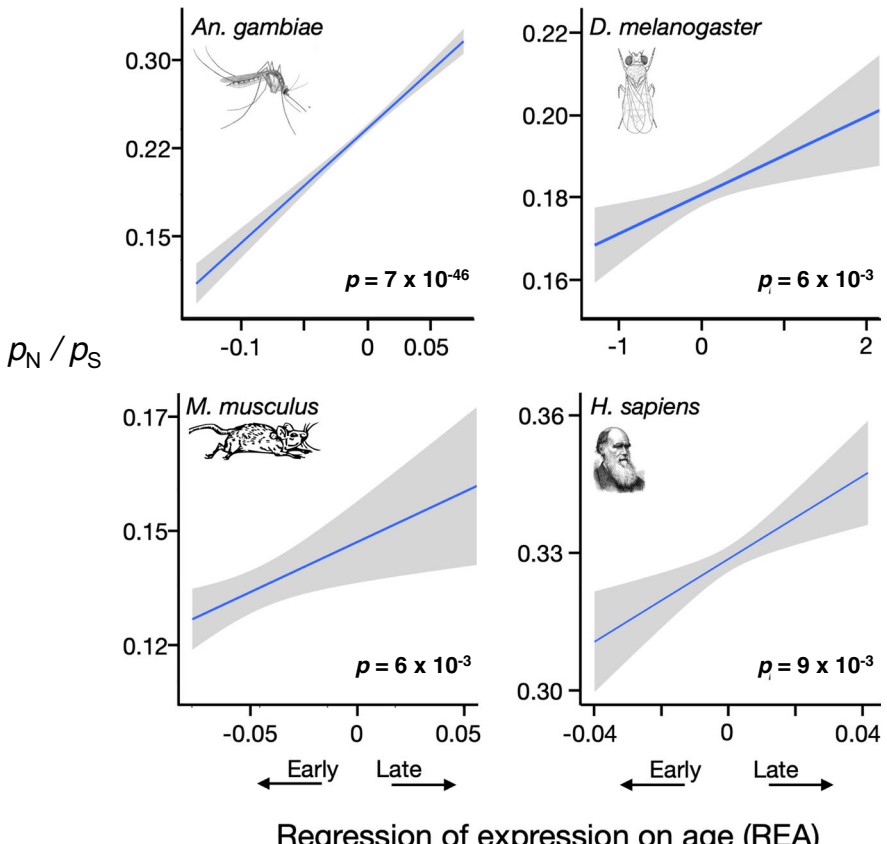

**Fig. 2 Late genes are more polymorphic for nonsynonymous mutations.** Genes that are more highly expressed late in life are significantly more polymorphic for nonsynonymous mutations than are genes expressed early in all four species. Nonsynonymous polymorphism is measured here as the fraction of sites in a gene that segregate for nonsynonymous alleles ($p_N$) relative to the fraction segregating for synonymous alleles ($p_S$). Significance was determined by Spearman rank correlations, and p-values in bold are significant at $p < 0.05$ (two-sided tests, not corrected for multiple tests). The lines are least squares regressions, and the gray regions show the approximate 95% confidence intervals for the regressions. See also Supplementary Fig. 2 and Supplementary Table 1.

synonymous sites. At the nonsynonymous sites, a fraction $\alpha_m$ of mutations are beneficial with a relative fitness effect as homozygotes of $s$ (>0), a fraction $1 - \alpha_m$ are deleterious with a fitness effect of $-s$, and heterozygotes have fitness intermediate between the two homozygotes. All mutations at synonymous sites are selectively neutral. We implemented the Poisson field model of molecular evolution[10] under these assumptions to obtain expressions for $d_N/d_S$ and $p_N/p_S$ that depend on just three quantities: $N_e s$, $\alpha_m$, and $\mu_N/\mu_S$. The latter quantity can be estimated directly from the relative numbers of nonsynonymous and synonymous sites in a gene. We then estimated $N_e s$ and $\alpha_m$ for each gene by fitting the model to the estimates of $d_N/d_S$ and $p_N/p_S$ (see "Methods" section).

The results are shown in Figs. 5 and 6, and Supplementary Figs. 5 and 6. The effective strength of selection ($N_e s$) declines with REA in all four species, significantly so in three of them. The fraction of mutations that are beneficial ($\alpha_m$) increases significantly with increasing values of REA in two of the four species. These results derive from a toy model, and the parameter values that it estimates cannot be taken literally. But the model suggests that the data are qualitatively consistent with declining effective population sizes, and consequently weaker effective selection, as life proceeds.

One might also expect that the distribution of fitness effects (DFE) of mutations that become fixed also varies with age of expression. We followed the approach of ref. [11] to estimate this distribution, but found no consistent relationship between the

DFE and REA. This is perhaps not surprising, however, in light of the results in the previous paragraph. If $N_e$ decreases with age but the fraction of mutations that are beneficial ($\alpha_m$) increases, the effects of age on the DFE of mutations that spread to fixation are likely to be complex.

**Discussion**

Purifying selection is one of the key processes that governs patterns of molecular variation[9]. The strength of purifying selection on coding regions, as reflected by polymorphism within and divergence between species, is known to be correlated with a number of factors. These include the local recombination rate, level of expression, breadth of expression, protein length, GC content, sex linkage, degree of sex-biased expression, and the location of a gene's product in a pathway (see[11–13] and the references therein). To that list we add the ages at which a gene is expressed.

Genes expressed late in life carry significantly more nonsynonymous polymorphism and are younger than genes expressed early in life. These correlations are consistent with a declining intensity of purifying selection with age, providing a link between life history theory and patterns of molecular evolution. The percentage of variance in these statistics that REA explains is not very large: for example, the greatest value for $r^2$ is 6% for $d_N/d_S$ in *An. gambiae* (Supplementary Table 3). This effect size is, however, equal or greater the effect of whether a gene is autosomal or sex-linked, which is considered to be a major determinant of

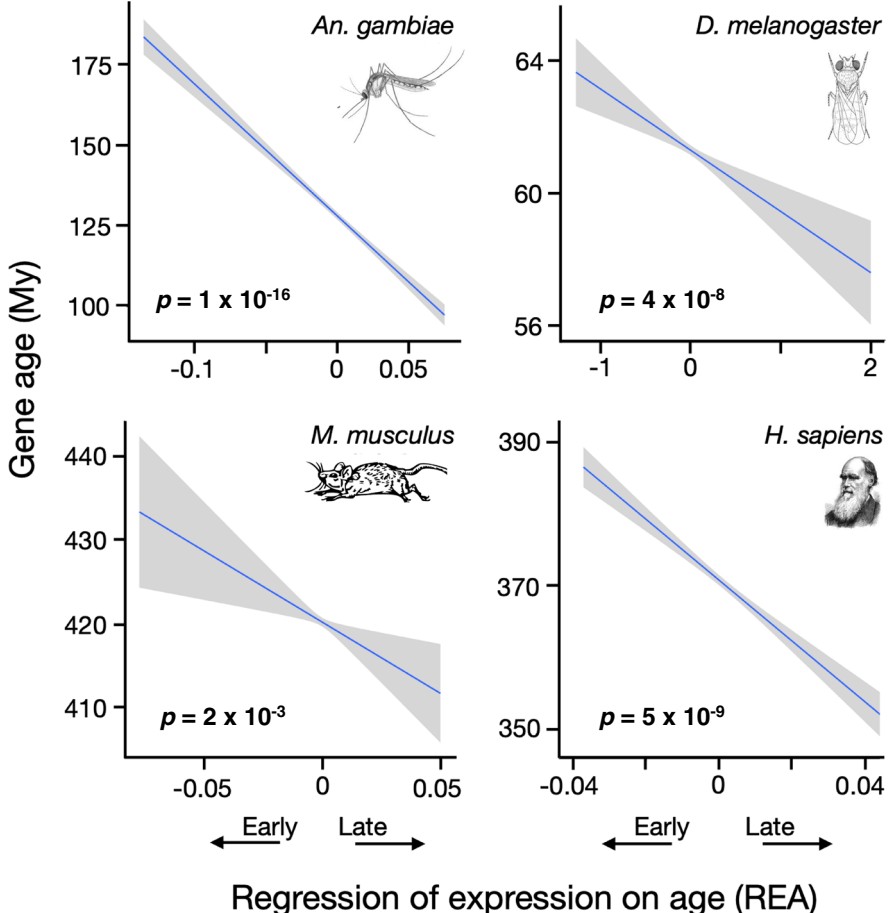

**Fig. 3 Late genes are younger.** Genes expressed later in life originated significantly more recently than those expressed early in all four species. Significance was determined by Spearman rank correlations, and p-values in bold are significant at $p < 0.05$ (two-sided tests, not corrected for multiple tests). The lines are least squares regressions, and the gray regions show the approximate 95% confidence intervals for the regressions. See also Supplementary Fig. 3 and Supplementary Table 1.

**Table 2 The effects of the ages at which pairs of paralogs are expressed on $p_N/p_S$, $d_N/d_S$, and gene age (in millions of years).**

| Species | N | $\Delta d_N/d_S$ | $\Delta p_N/p_S$ | $\Delta$Age |
|---------|-----|-------------|-------------|---------|
| Mosquito | 13,940 | 0.38*** | 0.99*** | −117*** |
| Fruit fly | 36,366 | 0.0055*** | 0.0068 | −4.0* |
| Mouse | 10,644 | 6.2*** | 2.7*** | −5300** |
| Human | 30,954 | 1.7*** | 1.2 | −177*** |

N is the number of pairs of paralogs compared. Entries in the table are the regressions of the difference in the statistic on the difference in REA for the pairs. Significance was determined using the Spearman rank correlations (two-sided tests, not corrected for multiple tests). Exact p values are given in Supplementary Table 1.
*$p < 0.05$, **$p < 0.01$, ***$p < 0.001$.

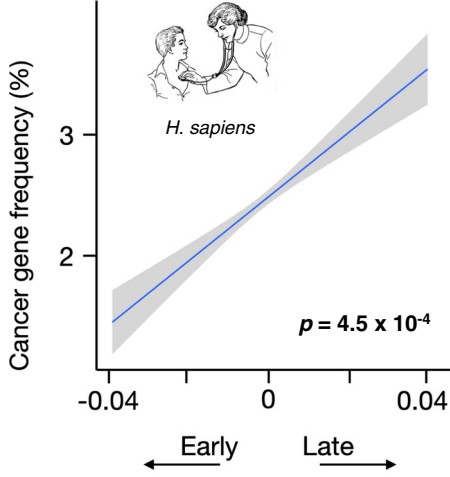

**Fig. 4 Late genes are more likely to be driver genes for adult cancers in humans.** The probability that a gene is associated with adult cancers is significantly correlated with its age of expression in humans. Significance was determined by a Spearman rank correlation (two-sided test). The line is the least squares regression, and the gray region shows the approximate 95% confidence interval for the regression. See also Supplementary Fig. 4.

patterns of molecular evolution[14]. Thus, the ages at which a gene is expressed appears to be a relatively important factor in determining patterns of molecular evolution.

Our interpretation of these patterns assumes there is a correlation between the age at which a gene is expressed and when selection acts on it. That correlation is certainly plausible for genes involved in processes such as cellular metabolism. The correlation could, however, break down for some types of genes. For example, early genes can give rise to structures that persist throughout life and so experience selection at all ages.

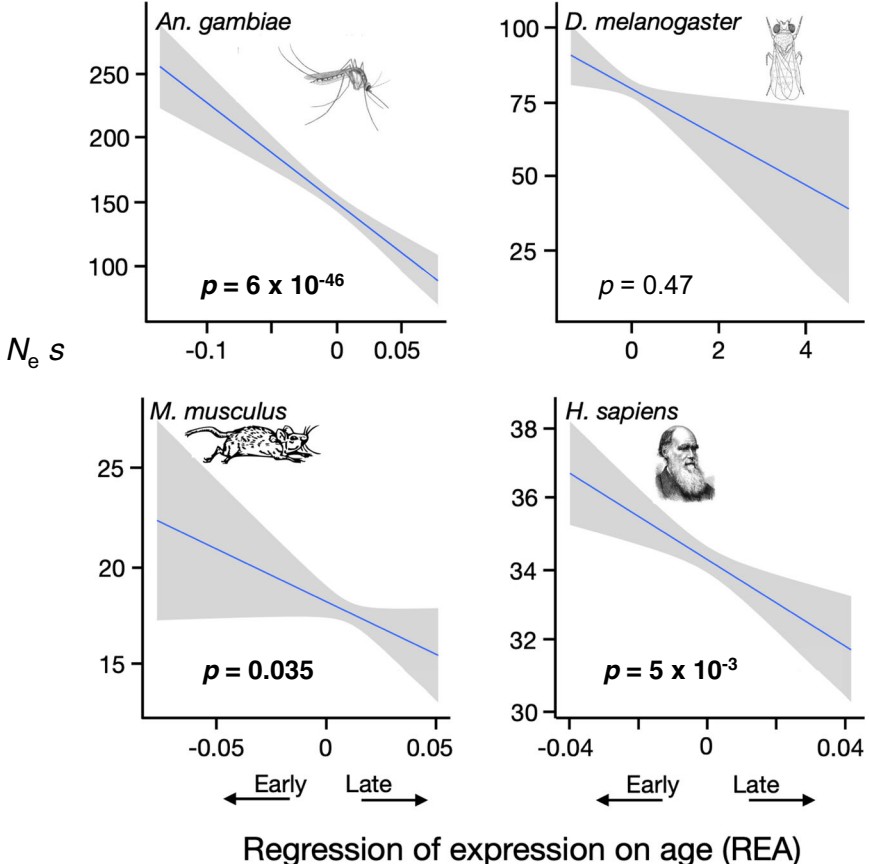

**Fig. 5 Selection on late genes is weaker.** The relative strength of selection is significantly weaker in late-expressed genes than early-expressed genes in three of the four species. The estimates of $N_e s$ are based on a highly simplified model that assumes beneficial mutations have fitness effect $s$ and deleterious mutations fitness $-s$. Significance was determined by Spearman rank correlations, and $p$-values in bold are significant at $p < 0.05$ (two-sided tests, not corrected for multiple tests). The lines are least squares regressions, and the gray regions show the approximate 95% confidence intervals for the regressions. See also Supplementary Fig. 5 and Supplementary Table 1.

Unfortunately, current methods for gene ontology do not allow this level of detail to be parsed.

The introduction reviewed studies that have previously reported some of the trends evident in our data. Some additional results are also consistent with ours. Campos et al.[11] found a negative relation between nonsynonymous divergence between species ($d_N$) and the strength of purifying selection (measured as $N_e s$ for deleterious mutations). Muntane et al.[15] and Kwalczyk et al.[16] found that across species of mammals, in some genes increased values of $d_N/d_S$ are correlated with life history traits such as longevity. Those results could result from differences in the intensity of purifying selection, as we suggest in this paper, but could also be explained by other factors correlated with longevity. Popadin et al.[17] reported that young eQTL (expression quantitative trait loci) that code for mRNA have higher $d_N/d_S$, consistent with our results for protein-coding regions. However, they interpreted the cause-and-effect differently: they proposed that genes become more constrained as they age, while we interpret gene age and polymorphism to both be influenced by age of expression. The hypotheses are not mutually exclusive.

An unexpected result is that the fraction of mutations that are beneficial ($\alpha_m$) increases with age of expression in all four species (Fig. 6). Why might that pattern emerge? It appears that late genes fix more deleterious mutations than early genes (Fig. 1). Thus they may have more opportunities for back mutations and compensatory mutations that increase fitness. To see the logic of that argument, consider a hypothetical gene under infinitely strong purifying selection. It would always be

fixed for the optimal nucleotide at every site and therefore never have the opportunity for compensatory or back mutations. In contrast, a gene under weaker purifying selection will occasionally fix a deleterious mutation, creating the possibility of a beneficial compensatory or back mutation.

Genes expressed at different ages also differ with respect to their cellular functions. Early genes are enriched in many gene ontology (GO) categories, and these show some consistency across species. In contrast, late genes only rarely show enrichment for any category (Supplementary Table 2). This difference could be explained if late genes are more often subject to misregulation as the result of accumulating more deleterious mutations. That idea is supported by our observation that late genes in humans are more than twice as likely to be driver genes for adult cancers than are early genes. This trend is likely the simple result of the fact that most cancers are late-onset, and therefore the genes that trigger those cancers will be expressed late in life. Fundamentally, the reason that cancers tend to be late-onset is because of the diminished strength of purifying selection at older ages. The same logic explains the observation that late genes more often segregate for dominant disease-causing mutations.

Our results are relevant to two general theories for the evolution of senescence: Medawar's mutation accumulation hypothesis and the antagonistic pleiotropy hypothesis[4,18]. Both postulate that senescence results from the accumulation of mutations with deleterious effects late in life. The difference between the two theories lies in the effects these mutations have early in life. Under the mutation accumulation hypothesis, mutations with

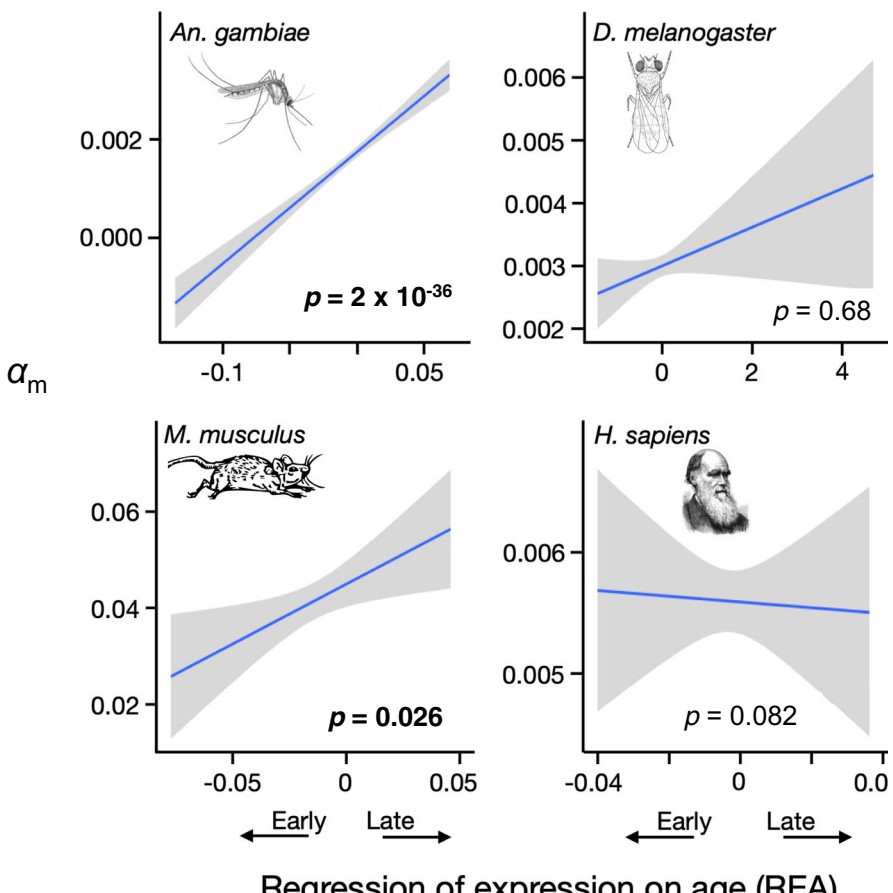

**Fig. 6 Late genes fix beneficial mutations more often.** The fraction of nonsynonymous mutations that are beneficial ($\alpha_m$) is significantly greater in late-expressed than early-expressed genes in two of the four species. Significance was determined by Spearman rank correlations, and $p$-values in bold are significant at $p < 0.05$ (two-sided tests, not corrected for multiple tests). The lines are least squares regressions, and the gray regions show the approximate 95% confidence intervals for the regressions. See also Supplementary Fig. 6 and Supplementary Table 1.

deleterious effects late in life are nearly neutral early in life, while under the antagonistic pleiotropy hypothesis they are beneficial early[18]. Both theories have found empirical support from quantitative genetic, molecular, and experimental studies of fitness-related traits (reviewed in the refs. [19,20]). The patterns seen here in $d_N/d_S$, $p_N/p_S$, and gene age are consistent with the mutation accumulation hypothesis, but are not necessarily expected under the antagonistic pleiotropy hypothesis.

In closing, we return to Medawar's hypothesis. We have focused on the molecular signatures that result from the decline in purifying selection with age. The increased accumulation of deleterious mutations in late-expressed genes evident in these phylogenetically diverse species suggests that those mutations may be contributing to their senescence, as Medawar first proposed 75 years ago.

## Methods

**Gene expression**. We collected data on expression of 12,332 genes in a malaria mosquito, *Anopheles gambiae*, using microarrays. Whole-body gene expression was measured on lab strains that were founded from a natural population in Cameroon, Central Africa. We extracted total RNA from three replicate pools of 20 adult females at five adult ages: 1, 2, 3, 4, and 5 weeks. To minimize the effects of circadian rhythm, we sampled at the same hour of the day following protocols described in the refs. [21,22] for RNA processing, raw data quality control, and filtering. The microarray data were analyzed using NimbleScan v2.5 (Roche NimbleGen).

For fruit flies (*D. melanogaster*), we analyzed two published datasets. First, we considered the expression of 12,703 genes in the whole bodies of 48 females sampled at young and old ages[23]. Those data were processed with the R packages *Bioconductor* and *oligo*[24,25]. Second, we analyzed expression of 15,139 genes in the

modENCODE dataset, which includes three males and three females at the ages of one day, five days, and 30 days[26,27].

In mice (*M. musculus*), we used the expression of 8932 genes in 20 females and 20 males at four ages in 16 tissues as reported in the Atlas of Gene Expression in Mouse Aging Project database[28]. In humans, we analyzed the expression of 39,962 genes in nine tissues from about 200 individuals in the Genotype-Tissue Expression (GTEx) dataset[29].

**Gene sequences**. To quantify molecular variation in natural populations of the malaria mosquito, we first used pooled whole genome sequencing of four populations from Cameroon[30]. Each pool contained 34 females, and average genome coverage was 45-fold. We followed ref. [30] for data quality control, filtering, mapping, and allele frequency estimates.

For fruit flies, we analyzed 621 genomes sampled from natural populations in the African ancestral range[31,32].

In mice, we used high-quality SNPs from the whole genome sequencing of wild-caught individuals[33]. The dataset includes about 50 individuals from seven populations from Europe and Asia.

For humans, we used allele frequencies estimated by the 1000 Genomes Project[34]. The sample includes more than 2000 individuals from 26 populations worldwide.

**Statistical analyses**. For the gene expression analyses, read counts were log transformed. Thus expression differences (e.g., between early-expressed and late-expressed genes) are scaled relative to overall expression levels.

For all four species, we quantified how gene expression changes with age using the regression coefficient of expression on age (REA). The REA values for the data we collected from the malaria mosquito are given in Supplementary Data 1. We also used the age when gene expression is maximum as an alternative way to quantify age-dependent gene expression. Results with the two methods are very similar, and so in the following we only report those using the REA.

To quantify $d_N/d_S$, for mosquitoes we used estimates from ref. [35], which are based on orthologous sequences from mosquitoes in the *An. gambiae* species

complex. For fruit flies, we used the estimates of $d_N/d_S$ provided by ref. [36]. For mice, we estimated $d_N/d_S$ between *M. musculus* and the rat *Rattus norvegicus* using their Ensembl genomes[37]. For humans, we used the $d_N/d_S$ ratios estimated for the human lineage reported by ref. [38].

To quantify the relative rate that $d_N/d_S$ changes with the age at which a gene is expressed, we defined the statistic $\Delta R_{NS}$:

$$\Delta R_{NS} = \frac{(d_N/d_S)_{Late} - (d_N/d_S)_{Early}}{\frac{1}{2}\left[(d_N/d_S)_{Late} + (d_N/d_S)_{Early}\right]}, \quad (1)$$

where $(d_N/d_S)_{Late}$ is the average value of $d_N/d_S$ among late-expressed genes, defined as those in the 90th percentile of REA values, and $(d_N/d_S)_{Early}$ is the average among early-expressed genes, defined as those in the 10th percentile of REA values.

For each species, we estimated $p_N/p_S$ for each gene in each population using *PopGenome*[39]. We then calculated the average of that ratio across all populations.

For the ages of genes in flies, we used estimates from ref. [40]. For the mouse and human, we used gene ages estimated by ref. [41]. To estimate the ages of genes in the malaria mosquito, we followed the approach used by ref. [40]. The origins of genes were dated by the presence or absence of orthologs on a phylogenetic tree that includes 18 species[35] (Supplementary Fig. 7). Genes that are specific to a single species are define here as young, since they originated since that species diverged from its closest relative. Genes shared by two or more species are defined here as old, since they are older than the most recent common ancestor of the youngest sister species. We retrieved the ortholog information from Vectorbase[42] using Biomart[43] and from OrthoDB[44]. The resulting estimates of gene age for the mosquito are given in Supplementary Data 1.

We evaluated how the age at which a gene is expressed affects patterns of molecular evolution using the regression model implemented in the R package *nlme*[45], where $y$ is the evolutionary statistic of interest, $x_{1i}$ is the REA of gene $i$, $x_{2i}$ is its mean expression level, and $e_i$ is residual error. In the two mammals, where expression was measured in multiple tissues, tissue was fit as a random factor. In the two insects, a gene's breadth of expression across tissues was fit as an additional factor using the specificity index of ref. [46].

$$y_i = b_0 + b_1 x_{1i} + b_2 x_{2i} + e_i \quad (2)$$

Regressions were fit separately to predict $p_N/p_S$, $d_N/d_S$, and gene age as the response variable ($y_i$). The significance of the REA coefficients ($b_1$) are shown in Figs. 1–3, where the $p$ values were determined using nonparametric Spearman rank correlation implemented in R (with no corrections for multiple tests). We also regressed $\pi_N/\pi_S$, the ratio of nonsynonymous to synonymous molecular diversity, onto REA and obtained results similar to those shown for $p_N/p_S$.

We compared statistics of molecular evolution in pairs of paralogs that are expressed at different ages (as measured by the REA statistic). Paralogue information was retrieved from the Ensembl genome database. Comparisons were made for all possible pairs of paralogs, with mean expression and tissue specificity as covariates. We regressed the differences in the values of $d_N/d_S$, $\Delta p_N/p_S$, and gene age for the early and late pairs of paralogs against the difference in their REA values. We denote these regression coefficients as $\Delta d_N/d_S$, $\Delta p_N/p_S$, and $\Delta$Age.

We asked whether genes that are expressed early and late are enriched for different gene ontology (GO) classes. GO subsets were downloaded from the Ensembl database. Gene set enrichment analysis was performed using R package *gage*[47]. To compare the GO terms enriched among early and late genes, we used the lower 10th percentile and upper 90th percentile of REA scores (as explained above following Eq. (1)).

We sought evidence that a gene's age of expression is prone to misregulation as reflected in the probability that it is a driver gene for adult cancers (i.e., linked to cancer progression when mutated). We compiled a list of 789 human adult cancer driver genes reported in pan-cancer studies[48–51]. Likewise, we asked if loci that segregate for dominant disease-causing mutations in humans are more frequent among late genes than early genes. We obtained a list of those loci from ref. [52]. We used nonparametric statistics (Spearman's rank correlation and the Wilcoxon rank sum test) to test for significant association between REA and the frequency of cancer driver genes and disease-causing genes.

**A model of gene lifespan**. We are interested in how the intensity of purifying selection affects the lifespan of a gene. Here, we present a highly simplified model for how a gene's lifespan is determined. The model is not intended to capture all the biological complexities, or to represent the only plausible factor that determines a gene's lifespan. The goal here is simply to show it is plausible that genes that are expressed later in life, and so are under weaker purifying selection, will tend to have shorter lifespans than genes expressed early.

When a gene originates, it is assigned a fitness of 1. We assume that a gene is lost when the fixation of deleterious mutations by drift causes its fitness to decline below a threshold that is less than 1. The loss of fitness by deleterious mutations is partly offset by the fixation of beneficial mutations. Homozygotes for a beneficial mutation have relative fitness $(1 + s)$, homozygotes for deleterious mutations have relative fitness $(1 − s)$, and heterozygotes have fitness intermediate between the homozygotes. We assume that mutations have additive effects, so the fixation of a mutation simply causes the gene's fitness to increase or decrease by an increment $s$, depending on whether the mutation is beneficial or deleterious. A fraction $\alpha$ of

mutations that become fixed are beneficial. The gene is lost when its fitness decreases to $1 − k s$, i.e., when $k$ more deleterious mutations than beneficial mutations have become fixed. The expected time to a gene's extinction, $\bar{t}$, is related to the expected total number of substitutions, $\bar{n}_T$, by

$$\bar{n}_T = \bar{t}(r_B + r_D), \quad (3)$$

where $r_B$ and $r_D$ are the fixation rates for beneficial and deleterious mutations.

To find $\bar{n}_T$, we model the evolution of fitness as a random walk. From Eq. XIV.4.12 in ref. [53], the generating function for the probability that the gene becomes extinct after a total of $n_T$ substitutions is

$$V(s) = \left\{ \frac{1 - [1 - 4\alpha(1 - \alpha)s^2]^{1/2}}{2\alpha s} \right\}^k. \quad (4)$$

We find the expected total number of substitutions before the gene becomes extinct by taking the derivative of $V(s)$ with respect to $s$ and then setting $s = 1$, which gives

$$\bar{n}_T = \frac{k}{2^k \sqrt{1 - 4\alpha(1 - \alpha)}} \left[ \frac{1 - \sqrt{1 - 4\alpha(1 - \alpha)}}{\alpha} \right]^k. \quad (5)$$

When very few of the substitutions are beneficial ($\alpha \ll 1$), we see that

$$\bar{n}_T \approx k(1 + 2\alpha) \approx k\left(1 + \frac{2r_B}{r_B + r_D}\right). \quad (6)$$

Substituting this result into Eq. (3) and rearranging then gives the expected time to a gene's extinction:

$$\bar{t} \approx \left(\frac{k}{r_B + r_D}\right)\left(1 + \frac{2r_B}{r_B + r_D}\right). \quad (7)$$

If we assume that the stronger purifying selection decreases the rate that deleterious mutations become fixed ($r_D$) but has no effect on the rate that beneficial mutations become fixed ($r_B$), then Eq. (7) shows that genes under weaker purifying selection (and therefore larger $r_D$) will become extinct at a younger age.

**A model to estimate $\alpha_m$ and $N_e s$.** To investigate how the ages at which a gene is expressed affect the fraction of nonsynonymous mutations ($\alpha_m$) that are beneficial and the strength of selection acting on them ($N_e s$), we developed a highly simplified model. The assumptions regarding mutation and selection are outlined in the main text. We incorporated those assumptions into the Poisson field model of Sawyer and Hartl[10], which in the following we refer to as SH. In brief, this model assumes that sites within a gene evolve without selective interference, either because they are in linkage equilibrium or because substitutions are dispersed in time. The model can be used to compute the probability density of allele frequencies and rates of fixation at sites segregating with alleles that have a given fitness effect. As SH emphasizes, this model is not equivalent to the infinite sites model.

We used this model to determine the relative frequencies of nonsynonymous and synonymous polymorphic sites, $p_N/p_S$, and the ratio of nonsynonymous to synonymous substitutions, $d_N/d_S$. The ratio of nonsynonymous to synonymous sites is

$$\frac{p_N}{p_S} = \frac{n_B + n_D}{n_S}, \quad (8)$$

where $n_B$, $n_D$, and $n_S$ are the expected numbers of polymorphic sites in a sample at which beneficial nonsynonymous mutations, deleterious nonsynonymous mutations, and neutral synonymous mutations occur. Using SH Eqs. (14) and (15) we find that for a sample of size $m$,

$$n_B = 4N_e \alpha_m \mu_N \int_0^1 \left\{ \left[ \frac{1 - \exp\{-4N_e s(1 - p)\}}{(1 - \exp\{-4N_e s\})p(1 - p)} \right] \left[1 - p^m - (1 - p)^m\right] \right\} dp, \quad (9)$$

$$n_S = 4N_e \mu_S \int_0^1 \left[ \frac{1 - p^m - (1 - p)^m}{p} \right] dp = 4N_e \mu_S \sum_{k=1}^{m-1} \frac{1}{k}, \quad (10)$$

where $N_e$ is the effective population size, $s$ is the fitness effect of a mutation, and $\mu_S$ and $\mu_N$ are the per-site rates of synonymous and nonsynonymous mutations. The value for $n_D$ is found by substituting $-s$ for $s$ in Eq. (9). Substituting these results into Eq. (8) shows that $p_N/p_S$ depends on just three quantities: $\alpha_m$, $N_e s$, and $\mu_N/\mu_S$. The integrals cannot be solved analytically, but the expression for $p_N/p_S$ can be evaluated numerically.

Regarding substitutions, we can write

$$\frac{d_N}{d_S} = \frac{F_B + F_D}{F_S}, \quad (11)$$

where $F_B$ and $F_D$ are the number of sites at which beneficial and deleterious nonsynonymous mutations become fixed in one species or the other, and $F_S$ is the number of synonymous mutations that have become fixed. Consider synonymous sites at which one allele is fixed in a sample of $m$ gene copies from a first species,

while an alternative allele is found in a sample of a single copy from a second species. From SH Eq. (17), we find that the expected number of such sites is

$$F_{\mathrm{S}} = 4N_{\mathrm{e}}\mu_{\mathrm{S}}\left(t_{\mathrm{div}} + \frac{1}{m} + 1\right), \qquad (12)$$

where $t_{\mathrm{div}}$ is the time since the two species diverged, measured in units of $2N_{\mathrm{e}}$ generations. The corresponding numbers of sites fixed for beneficial and deleterious nonsynonymous mutations are

$$F_{\mathrm{B}} = 4N_{\mathrm{e}}\alpha_{\mathrm{m}}\mu_{\mathrm{N}}\left(\frac{N_{\mathrm{e}}s}{1 - \exp\{-N_{\mathrm{e}}s\}}\right)\left[t_{\mathrm{div}} + G_{\mathrm{B}}(m) + G_{\mathrm{B}}(1)\right], \qquad (13)$$

$$F_{\mathrm{D}} = 4N_{\mathrm{e}}(1 - \alpha_{\mathrm{m}})\mu_{\mathrm{N}}\left(\frac{-N_{\mathrm{e}}s}{1 - \exp\{N_{\mathrm{e}}s\}}\right)\left[t_{\mathrm{div}} + G_{\mathrm{D}}(m) + G_{\mathrm{D}}(1)\right], \qquad (14)$$

where

$$G_{\mathrm{B}}(n) = \int_0^1 x^{n-1}\left[\frac{1 - \exp\{-N_{\mathrm{e}}s(1-x)\}}{N_{\mathrm{e}}s(1-x)}\right]\mathrm{d}x. \qquad (15)$$

$G_{\mathrm{D}}(n)$ is found by replacing $s$ with $-s$ in Eq. (15). Substituting these expressions into Eq. (11) gives $d_{\mathrm{N}}/d_{\mathrm{S}}$. As with $p_{\mathrm{N}}/p_{\mathrm{S}}$, the result depends only on $\alpha_{\mathrm{m}}$, $N_{\mathrm{e}}s$, and $\mu_{\mathrm{N}}/\mu_{\mathrm{S}}$. When the species have diverged much more than $N_{\mathrm{e}}$ generations ago, the result simplifies to

$$\frac{d_{\mathrm{N}}}{d_{\mathrm{S}}} \approx \frac{\mu_{\mathrm{N}}}{\mu_{\mathrm{S}}}\left[\alpha_{\mathrm{m}}\left(\frac{N_{\mathrm{e}}s}{1 - \exp\{-N_{\mathrm{e}}s\}}\right) + (1 - \alpha_{\mathrm{m}})\left(\frac{-N_{\mathrm{e}}s}{1 - \exp\{N_{\mathrm{e}}s\}}\right)\right]. \qquad (16)$$

To fit this model to the data, we estimated $\mu_{\mathrm{N}}/\mu_{\mathrm{S}}$ as the ratio of the numbers of nonsynonymous and synonymous sites in a gene. Then the expressions for $p_{\mathrm{N}}/p_{\mathrm{S}}$ and $d_{\mathrm{N}}/d_{\mathrm{S}}$ depend on the only two remaining parameters, $N_{\mathrm{e}}s$ and $\alpha_{\mathrm{m}}$. Given the estimates of $p_{\mathrm{N}}/p_{\mathrm{S}}$ and $d_{\mathrm{N}}/d_{\mathrm{S}}$ for a gene, we solved Eqs. (8) and (16) numerically to obtain estimates of $N_{\mathrm{e}}s$ and $\alpha_{\mathrm{m}}$. We do not know the statistical properties of these estimators (for example, if they are biased), but we have no reason to believe those properties could lead to a spurious correlations between REA and estimates of $N_{\mathrm{e}}s$ and $\alpha_{\mathrm{m}}$.

**Reporting summary**. Further information on research design is available in the Nature Research Reporting Summary linked to this article.

## Data availability

Our gene expression data for *An. gambiae* have been deposited in the ArrayExpress database under accession numbers E-MTAB-10110 and E-MTAB-2463, in the N.C.B.I. Gene Expression Omnibus under accession number GSE18194, and in the *Anopheles gambiae* Gene Expression Database (http://www.angagepuci.bio.uci.edu/). Gene expression data for *D. melanogaster* are from modEncode (http://data.modencode.org/?Organism=D.%20melanogaster). Gene expression data for *M. musculus* are from the Atlas of Gene Expression in Mouse Aging Project (http://cmgm.stanford.edu/~kimlab/aging_mouse/). Data on gene expression in humans came from GTEx (https://www.gtexportal.org/home/). Genome sequence and variants for mosquitoes came from the *Anopheles gambiae* 1000 Genomes Project (http://www.malariagen.net/ag1000g), for flies from the *Drosophila* Genome Nexus (http://www.johnpool.net/genomes.html), for wild populations of mouse from the GWGD repository (http://wwwuser.gwdg.de/~evolbio/evolgen/wildmouse/), and for human from the 1000 Genomes Project (https://www.internationalgenome.org/). To estimate gene ages in *An. gambiae*, we used data on orthologues from Vectorbase (https://vectorbase.org/vectorbase/) and OrthoDB (https://www.orthodb.org). Gene paralogues were identified using Ensembl (http://www.ensembl.org/). Gene ontology (GO) classes were determined using the Ensembl genome database (http://www.ensembl.org/) and the Database for Annotation, Visualization, and Integrated Discovery (https://david.ncifcrf.gov/).

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

## Acknowledgements

C.C. thanks Prof. Nora Besansky for her long-term mentorship and intellectual support. We thank D. Houle, V. Narasimhan, J.M. Sardell, and M. Slatkin for discussions. This research was supported by NIH grant R01 GM116853 to M.K.

## Author contributions

C.C. and M.K. collaborated on all aspects of this project.

## Competing interests

The authors declare no competing interests.
