## [Peer Review File · Nature Communications]

Reviewers' Comments:

Reviewer #1:

Remarks to the Author:

Review of Cheng & Kirkpatrick

This study uses a simple and clever (and thus a very elegant) approach to examine the relationship between a gene's age-dependent expression level, its age, and two metrics of molecular evolution, pn/ps (the ratio of non-synonymous to synonymous polymorphisms) and dn/ds (the ratio of non-synonymous to synonymous substitutions). Consistent with the notion that the force (intensity) of (purifying) selection declines with age, the authors report that genes which are expressed late in life (relative to those expressed early in life) are more polymorphic for nonsynonymous (potentially deleterious) mutations, fix nonsynonymous mutations more frequently and have evolutionary shorter lifespans. The authors find that these patterns hold for *Homo sapiens*, *Mus musculus*, *Anopheles gambiae*, and *Drosophila melanogaster*, thus suggesting that they are general. In contrast to early-expressed genes, late-expressed genes exhibit no consistent gene ontology enrichment, again potentially consistent with the idea of weakened purifying selection at advanced age. Moreover, in humans, late-expressed genes are more likely to be linked to cancer.

The paper is very clearly and succinctly written; and the methods are appropriate and clear. I also liked the clear and self-critical discussion of the findings.

In my opinion, this study provides an elegant and robust confirmation of Medawar's 1952 mutation accumulation hypothesis for the evolution of aging. More generally, the study provides a fruitful but currently still underexplored connection between patterns of molecular evolution, age-dependent gene expression and life-history evolution; it will clearly be of great interest to apply the approach taken here to additional datasets from other species in future work. For example, beyond expanding the scope of the present analyses to other taxa, it might be very interesting to apply the authors' approach to data from mutation accumulation lines and/or longevity selection lines, e.g. in *Drosophila*.

I really enjoyed reading this very interesting and well-written paper; I do not have any substantial concerns or comments.

Extremely minor comments:

- P1, keywords: maybe add aging?
- P2, abstract: consider briefly mentioning evolution of aging and mutation accumulation hypothesis?
- P3, Intro, L36: cite mutation accumulation mechanism, i.e. Medawar PB. 1952. *An Unsolved Problem of Biology*. London: H.K. Lewis.
- P3, Intro: more generally, very briefly mention evolution of senescence?

Reviewer #2:

Remarks to the Author:

This paper is a nice addition to the growing body empirical literature that shows that the decreasing strength of natural selection with age can be detected in our genomes. Additionally, it shows some of the consequences in terms of evolution and human health. Some of the results were expected (and, as I will mention below, already explored by others using different methods), but there is a good amount of very nice and innovative results. For instance, the suggestion that late genes fix more deleterious mutations than early genes and that, thus, they may have more opportunities for fitness-increasing compensatory mutations has beautiful implications for the genetic architecture of late-onset

disease.

Overall, the analysis is very thorough, the results seem very solid, the paper is very well written and the display items are quite sleek. As to the model: even if it is quite simple (as the authors themselves acknowledge) it is totally sound and very easy to follow, plus it does it work perfectly well. Overall, this is an interesting paper that makes a very convincing case that mutation accumulation is a general senescence mechanism.

***My main major comment is that a better contextualization work is necessary and that linking the paper with previous literature on the matter will make it far more visible. For instance:

- The idea that late genes (relatively highly expressed in older individuals) are under weaker purifying selection than early genes (genes relatively highly expressed in younger individuals) has already been tested by Turan et al. in 2019 (PMID: 31062469). Some of the results in this manuscript, particularly those related to dN/dS, were already present in that paper.

- A version of this idea, focusing on disease association rather than on gene expression, was also tested by Rodriguez et al. in 2017 (PMID: 28812720), who showed that SNPs associated to late-onset disease show higher risk allele frequency than SNPs associated to early onset disease and that, thus, genes associated to late-onset diseases may be more polymorphic.

- Back to rates of protein evolution, they have been linked to senescence by several authors, even if here the relation is not so direct. Two papers that come to mind are Kowalczyk et al. 2020 (PMID: 32043462) and Muntané et al 2018 (PMID: 29788292). In both cases, they test accelerated rates of protein evolution and correlate them with longevity across mammalian and primate species.

***Another point is that it would be much better to see versions of the figures with the data points, instead of the current versions that only show trends and confidence intervals.

***As to minor comments. I only have three:

- The idea that deleterious mutations are fixed more frequently in late genes can be tested using codon bias. It would be a nice addition, but I would be happy if the authors decide not to do it, since it would add little to their main points.

- As the authors mention, their interpretation of their observations depends on the assumption that there is a correlation between the age at which a gene is expressed and when it manifests a phenotypic effect such that selection, even if weaker, can act upon it. It would be nice to see if this holds in the context of the enormous literature on the matter. I think it does, but I would encourage the authors to be a bit more explicit.

- Finally, in lines 104-106 the statement is made that "In all four species, pN/pS and dN/dS are significantly greater, and gene age is significantly younger, in genes that are expressed early compared to their paralogs that are expressed late (Table 2)." That is a mistake, and Table 2 shows exactly the opposite trend, which is of course consistent with the whole paper.

Reviewer #3:

Remarks to the Author:

This paper examines the relationship between timing of expression of genes during the life cycle and metrics for the strength of selection on those genes. Classical population genetic theory says that the efficiency of natural selection should decline with age, and the paper largely consists of showing that this appears to be true in many different guises. It is surprising that this has not been done in this genome-

wide way, and it is such a sensible question to ask, the reader is naturally drawn into the question. The paper has some key strengths – the analysis was done with four distinct species – human, mouse, *Drosophila* and mosquito – and results are largely consistent, although the smaller effective size of humans makes selection appear generally weaker. The idea of looking at duplicated genes, where the two paralogs are expressed early vs late, is a nice addition. There is an appealing simplicity to just go for the Regression of Expression on Age. But it also raises some flags that probably warrant a bit more double checking. Here are some specific comments:

1. The role of tissue specificity of gene expression is glossed over too quickly. It is probably the case that early expressed genes are expressed in more tissues (more ubiquitously), and so one could argue that the observed patterns (REA vs dn/ds , for example) are driven by the fact that genes that are more tissue specific have weaker selection acting on them. This could easily be tested with the human and mouse data, which include multiple tissues. One could repeat the analysis stratified by tau, or some other metric for tissue specificity.
2. An awful lot rides on REA, and this appears to have been done by simple linear regression. While one wants to applaud simple approaches, linear regression can do funny things if the distributions are badly skewed etc.. (And gene expression generally has a long tail to the right). So let's do some Statistics 101 testing here – are the residuals normally distributed? Homoscedastic? Show a bit more detail of these quality checks in the supplement. Not seeing any plots at all with the real data is a bit unnerving. Some of the raw scatterplots should be in the supplement.
3. The argument of lines 184-187 is truly odd. This part reads: "An unexpected result is that the fraction of mutations that are beneficial (a_m) increases with age of expression in the mosquito (Fig. 5). Why might that pattern emerge? It appears that late genes fix more deleterious mutations than early genes (Fig. 2). Thus they may have more opportunities for back mutations and compensatory mutations that increase fitness." So the fraction of favorable mutations is elevated by the fact that there are more deleterious mutations that need compensation? Really?
4. There may be some circularity to the argument that late genes are more likely to be cancer drivers. Or at any rate, it is important to note that most cancers are late onset, so the associations with cancer are more likely to be with genes that are expressed when the cancer is seen.
5. Around lines 304, the argument gets stretched in funny ways when one lumps together genes that may or may not be essential. The objective here is to show that genes that are expressed later in life, (and so are under weaker purifying selection), will tend to have shorter lifespans than genes that are expressed early. But all else being equal, the probability that a gene is essential to life and reproduction is surely higher for early expressed genes. I suppose this is just the converse of saying late genes are under weaker selection, but it makes the exercise less exciting to realize that you are testing whether essential genes disappear at a slower rate than non-essential genes.
6. The Poisson Random Field modeling may need some additional work. While the algebra to get expressions that relate p_n/p_s , dn/ds , $N_e(s)$ and α appear to be okay, simple substitution may not produce statistical estimators of much value. The reader needs to be convinced that the estimators have nice properties like consistency, low bias and low variance. Estimates using ratios of terms (like dn/ds) seem especially likely to cause problems.

Reply to the reviews of NCOMMS 20-42575

We thank the three reviewers for their many constructive suggestions. We have incorporated almost all of them in the revision. The details follow.

Review 1

This study uses a simple and clever (and thus a very elegant) approach to examine the relationship between a gene's age-dependent expression level, its age, and two metrics of molecular evolution, pn/ps (the ratio of non-synonymous to synonymous polymorphisms) and dn/ds (the ratio of non-synonymous to synonymous substitutions). Consistent with the notion that the force (intensity) of (purifying) selection declines with age, the authors report that genes which are expressed late in life (relative to those expressed early in life) are more polymorphic for nonsynonymous (potentially deleterious) mutations, fix nonsynonymous mutations more frequently and have evolutionary shorter lifespans. The authors find that these patterns hold for *Homo sapiens*, *Mus musculus*, *Anopheles gambiae*, and *Drosophila melanogaster*, thus suggesting that they are general. In contrast to early-expressed genes, late-expressed genes exhibit no consistent gene ontology enrichment, again potentially consistent with the idea of weakened purifying selection at advanced age. Moreover, in humans, late-expressed genes are more likely to be linked to cancer.

The paper is very clearly and succinctly written; and the methods are appropriate and clear. I also liked the clear and self-critical discussion of the findings.

In my opinion, this study provides an elegant and robust confirmation of Medawar's 1952 mutation accumulation hypothesis for the evolution of aging. More generally, the study provides a fruitful but currently still underexplored connection between patterns of molecular evolution, age-dependent gene expression and life-history evolution; it will clearly be of great interest to apply the approach taken here to additional datasets from other species in future work. For example, beyond expanding the scope of the present analyses to other taxa, it might be very interesting to apply the authors' approach to data from mutation accumulation lines and/or longevity selection lines, e.g. in *Drosophila*.

I really enjoyed reading this very interesting and well-written paper; I do not have any substantial concerns or comments.

Extremely minor comments:

- P1, keywords: maybe add aging?

We added "aging" to the keywords.

- P2, abstract: consider briefly mentioning evolution of aging and mutation accumulation hypothesis?

We added "evolution of senescence" and "the accumulation of deleterious mutations" to the abstract (ll. 33-34).

- P3, Intro, L36: cite mutation accumulation mechanism, i.e. Medawar PB. 1952. An Unsolved Problem of Biology. London: H.K. Lewis.

We added the suggested references to Medawar's mutation accumulation hypothesis (ll. 41-42, 50, 225) and the new final paragraph of the main text starting at line 235. These changes definitely improve the paper.

- P3, Intro: more generally, very briefly mention evolution of senescence?

The paper now talks about senescence in the abstract (l. 32), Introduction (l. 42), and Discussion (ll. 225-227, 238).

Review 2

This paper is a nice addition to the growing body empirical literature that shows that the decreasing strength of natural selection with age can be detected in our genomes. Additionally, it shows some of the consequences in terms of evolution and human health. Some of the results were expected (and, as I will mention below, already explored by others using different methods), but there is a good amount of very nice and innovative results. For instance, the suggestion that late genes fix more deleterious mutations than early genes and that, thus, they may have more opportunities for fitness-increasing compensatory mutations has beautiful implications for the genetic architecture of late-onset disease.

Overall, the analysis is very thorough, the results seem very solid, the paper is very well written and the display items are quite sleek. As to the model: even if it is quite simple (as the authors themselves acknowledge) it is totally sound and very easy to follow, plus it does it work perfectly well. Overall, this is an interesting paper that makes a very convincing case that mutation accumulation is a general senescence mechanism.

***My main major comment is that a better contextualization work is necessary and that linking the paper with previous literature on the matter will make it far more visible. For instance:

- The idea that late genes (relatively highly expressed in older individuals) are under weaker purifying selection than early genes (genes relatively highly expressed in younger individuals) has already been tested by Turan et al. in 2019 (PMID: 31062469). Some of the results in this manuscript, particularly those related to dN/dS, were already present in that paper.

- A version of this idea, focusing on disease association rather than on gene expression, was also tested by Rodriguez et al. in 2017 (PMID: 28812720), who showed that SNPs associated to late-onset disease show higher risk allele frequency than SNPs associated to early onset disease and that, thus, genes associated to late-onset diseases may be more polymorphic.

- Back to rates of protein evolution, they have been linked to senescence by several authors, even if here the relation is not so direct. Two papers that come to mind are Kowalczyk et al. 2020 (PMID: 32043462) and Muntané et al 2018 (PMID: 29788292). In both cases, they test accelerated rates of protein evolution and correlate them with longevity across mammalian and primate species.

We are very grateful to the reviewer for pointing us towards literature that we didn't know about (but should have). We added the suggested references and others at lines 43-50 and 193-204.

***Another point is that it would be much better to see versions of the figures with the data points, instead of the current versions that only show trends and confidence intervals.

We added supplemental figures (S1 – S6) that show the individual points. We have kept the original plots in the main text because they show the overall trends much more clearly.

***As to minor comments. I only have three:

- The idea that deleterious mutations are fixed more frequently in late genes can be tested using codon bias. It would be a nice addition, but I would be happy if the authors decide not to do it, since it would add little to their main points.

Thanks for the suggestion of looking at codon bias. We did so and found inconsistent results. We decided not to include those results simply because they are not useful.

- As the authors mention, their interpretation of their observations depends on the assumption that there is a correlation between the age at which a gene is expressed and when it manifests a phenotypic effect such that selection, even if weaker, can act upon it. It would be nice to see if this holds in the context of the enormous literature on the matter. I think it does, but I would encourage the authors to be a bit more explicit.

We are uncertain about what literature the reviewer has in mind regarding the connection between when a gene is expressed and when selection acts on it.

Finally, in lines 104-106 the statement is made that “In all four species, pN/pS and dN/dS are significantly greater, and gene age is significantly younger, in genes that are expressed early compared to their paralogs that are expressed late (Table 2).” That is a mistake, and Table 2 show exactly the opposite trend, which is of course consistent with the whole paper.

Thanks for catching the interchanged “early” and “late” in that sentence. The mistake has been fixed (l. 121-122).

Review 3

This paper examines the relationship between timing of expression of genes during the life cycle and metrics for the strength of selection on those genes. Classical population genetic theory says that the efficiency of natural selection should decline with age, and the paper largely consists of showing that this appears to be true in many different guises. It is surprising that this has not done in this genome-wide way, and it is such a sensible question to ask, the reader is naturally drawn into the question. The paper has some key strengths – the analysis was done with four distinct species – human, mouse, *Drosophila* and mosquito – and results are largely consistent, although the smaller effective size of humans makes selection appear generally weaker. The idea of looking at duplicated genes, where the two paralogs are expressed early vs late, is a nice addition. There is an appealing simplicity to just go for the Regression of Expression on Age. But it also raises some flags that probably warrant a bit more double checking. Here are some specific comments:

1. The role of tissue specificity of gene expression is glossed over too quickly. It is probably the case that early expressed genes are expressed in more tissues (more ubiquitously), and so one could argue that the observed patterns (REA vs dn/ds , for example) are driven by the fact that genes that are more tissue specific have weaker selection acting on them. This could easily be tested with the human and mouse data, which include multiple tissues. One could repeat the analysis stratified by tau, or some other metric for tissue specificity.

We agree that controlling for the breadth of a gene's expression is important, and we have done that. Please see lines 73-74, 120, and 315-318.

2. An awful lot rides on REA, and this appears to have been done by simple linear regression. While one wants to applaud simple approaches, linear regression can do funny things if the distributions are badly skewed etc.. (And gene expression generally has a long tail to the right). So let's do some Statistics 101 testing here – are the residuals normally distributed? Homoscedastic? Show a bit more detail of these quality checks in the supplement. Not seeing any plots at all with the real data is a bit unnerving. Some of the raw scatterplots should be in the supplement.

We thank the reviewer for pointing out that parametric significance tests based on linear regression are inappropriate here. We have replaced those with p values computed from nonparametric Spearman rank correlations and Wilcoxon rank sum tests (ll. 77, 143, 321, 343-344; Table 2; Fig 1). As noted above, we've added supplemental figures to show each of the regressions plotted with the data points.

3. The argument of lines 184-187 is truly odd. This part reads: “An unexpected result is that the fraction of mutations that are beneficial (am) increases with age of expression in the mosquito (Fig. 5). Why might that pattern emerge? It appears that late genes fix more deleterious mutations than early genes (Fig. 2). Thus they may have more opportunities for back mutations and compensatory mutations that increase fitness.” So the fraction of favorable mutations is elevated by the fact that there are more deleterious mutations that need compensation? Really?

We believe that our argument is valid, but obviously our logic wasn't clear. We have added an explanation at lines 205-213. To summarize the idea: a gene that is under extremely strong purifying selection will always be fixed for the optimum sequence and so can never fix beneficial mutations. In contrast, a gene under weak purifying selection will sometimes fix a deleterious mutation, providing the opportunity for beneficial compensatory or back mutations to fix.

4. There may be some circularity to the argument that late genes are more likely to be cancer drivers. Or at any rate, it is important to note that most cancers are late onset, so the associations with cancer are more likely to be with genes that are expressed when the cancer is seen.

We agree with the reviewer, and again the problem seems to be with our lack of clarity. Essentially our point is that expression of driver mutations and cancers themselves are largely late in life because of diminished strength of selection. This is not at all a novel idea, but we feel that it's useful to show that our statistical framework gives sensible results in this context. We have added sentences at lines 220-224 that we hope clarify the meaning. We have also added a new finding: late genes are enriched of dominant human disease genes, while early genes have a greater proportion of recessive disease genes. This is again consistent with the idea that the strength of purifying selection diminishes over age.

5. Around lines 304, the argument gets stretched in funny ways when one lumps together genes that may or may not be essential. The objective here is to show that genes that are expressed later in life, (and so are under

weaker purifying selection), will tend to have shorter lifespans than genes that are expressed early. But all else being equal, the probability that a gene is essential to life and reproduction is surely higher for early expressed genes. I suppose this is just the converse of saying late genes are under weaker selection, but it makes the exercise less exciting to realize that you are testing whether essential genes disappear at a slower rate than non-essential genes.

The reviewer feels that it is uninteresting that early-expressed genes, which may more often be essential, are less likely to be lost than late-expressed genes. We agree that this result has a simple interpretation in retrospect, but it is not a generally known fact. In any event, the other two reviewer (and we ourselves) do find this result of interest.

6. The Poisson Random Field modeling may need some additional work. While the algebra to get expressions that relate p_n/p_s , dn/ds , $N_e(s)$ and α appear to be okay, simple substitution may not produce statistical estimators of much value. The reader needs to be convinced that the estimators have nice properties like consistency, low bias and low variance. Estimates using ratios of terms (like dn/ds) seem especially likely to cause problems.

The reviewer is correct that we do not know the statistical properties of the estimators that we devised. They are, however, the only estimators available. Further, we do not see how bias or other properties that they might have could lead to the correlations we observe between the ages at which genes are expressed and our estimates of N_e s and α_m . The revision makes these points at lines 428-431.

Reviewers' Comments:

Reviewer #2:

Remarks to the Author:

The authors have been very thorough with the reviewers' comments and have added new angles that make this manuscript even more enjoyable than it was to begin with. This is a great paper and will be, one hopes, very influential.

Reviewer #3:

Remarks to the Author:

The authors did a thorough job in revising the manuscript, and I am satisfied with the revisions. In my view the paper is now ready for publication in Nature Communications.